# The Beneficial Effects of Regular Intake of *Lactobacillus paragasseri* OLL2716 on Gastric Discomfort in Healthy Adults: A Randomized, Double-Blind, Placebo-Controlled Study

**DOI:** 10.3390/nu16183188

**Published:** 2024-09-20

**Authors:** Naruomi Yamada, Kyosuke Kobayashi, Akika Nagira, Takayuki Toshimitsu, Asako Sato, Hiroshi Kano, Kenichi Hojo

**Affiliations:** 1Health Science Research Unit, Division of Research and Development, Meiji Co., Ltd., Tokyo 192-0919, Japan; naruomi.yamada@meiji.com (N.Y.); akika.nagira.aa@meiji.com (A.N.); takayuki.toshimitsu@meiji.com (T.T.); asako.satou@meiji.com (A.S.); 2Wellness Science Labs, Meiji Holdings Co., Ltd., Tokyo 192-0919, Japan; kyousuke.kobayashi@meiji.com (K.K.); hiroshi.kano@meiji.com (H.K.)

**Keywords:** probiotics, *Lactobacillus paragasseri* OLL2716, gastrointestinal discomfort, autonomic nervous system, mental stress, appetite

## Abstract

We investigated the effects of *Lactobacillus paragasseri* OLL2716 on gastrointestinal symptoms in healthy adults with gastric complaints. In this randomized, double-blind, placebo-controlled trial, 174 healthy Japanese adults were randomly assigned to an OLL2716 or placebo group, and each group consumed 85 g of yogurt containing *L. paragasseri* OLL2716 or placebo yogurt daily for 12 weeks. The primary endpoint was the change in gastric symptoms from baseline as per the participants’ questionnaires at 6 and 12 weeks. The secondary endpoints were changes from baseline in the short-form Nepean Dyspepsia Index (SF-NDI), the Gastrointestinal Symptom Rating Scale (GSRS), and the Council on Nutrition Appetite Questionnaire-Japanese (CNAQ-J) scores at 6 and 12 weeks. The primary endpoint data showed that the changes in “epigastric pain” at 6 and 12 weeks were significantly decreased in the OLL2716 group compared with those in the placebo group. Additionally, the changes in “epigastric pain syndrome-like symptoms” were significantly decreased in the OLL2716 group compared with those in the placebo group at 6 weeks. The SF-NDI items that improved at 6 weeks were “irritable, tense, or frustrated”, “enjoyment of eating or drinking”, and “tension”, which are sub-scales related to mental stress. The items “Over-all” in the GSRS and “feeling hungry” in the CNAQ-J significantly improved in the OLL2716 group compared with the placebo group at 12 weeks. The results suggest that regular intake of *L. paragasseri* OLL2716 may improve both gastric discomfort and mental stress in healthy adults with gastric complaints, such as postprandial fullness or early satiety.

## 1. Introduction

Mental stress in daily life is thought to be related to gastrointestinal dysfunction and subsequent harmful symptoms because it disrupts the autonomic nervous system. Several studies have suggested that disturbances in the autonomic nervous system can cause delayed gastric motility [1,2,3]. A recent systematic review and meta-analysis supported the association between optimally measured delayed gastric emptying and upper gastrointestinal symptoms [4]. Furthermore, several studies concerning the stomach–brain axis have been performed, and they have reported the existence of a mutual communication between the brain and stomach via the autonomic nervous system [5,6,7].

Mental stress affects gastric motility, visceral perception, and the secretion of gastric hormone [8]. In contrast, stomach dysfunction, such as stomach pain and overloaded stomach acid, causes mental stress because it directly impairs the quality of life [9,10]. According to epidemiological studies, 50% of functional gastroenteropathies are caused by psychological stress, whereas the remaining 50% are caused by gastrointestinal tract dysfunction [11,12,13]. Although the cause-and-effect relationship remains unclear [14], maintaining both mental health and normal stomach function is essential.

Recently, several probiotic strains have been reported to exert beneficial effects on gastrointestinal function and discomfort [3,15,16]. *Lactobacillus paragasseri* OLL2716 (formerly known as *Lactobacillus gasseri*) has been reported to have beneficial effects, such as suppressive effects on *Helicobacter pylori* infection and the improvement of functional dyspepsia (FD), which is a condition with persistent abdominal symptoms mainly affecting the epigastrium [17,18]. Otomi et al. reported that *L. paragasseri* OLL2716 may ameliorate autonomic nervous system disorders [15]. Ohtsu et al. reported that *L. paragasseri* OLL2716 improved delayed gastric emptying and salivary amylase concentrations in healthy adults with stomach dysfunction [3]. In this study, we investigated the effects of *L. paragasseri* OLL2716 on gastric discomfort and mental stress in healthy adults with gastric complaints.

## 2. Materials and Methods

### 2.1. Study Design

A randomized, double-blind, placebo-controlled, parallel-group trial was conducted by assigning participants to either the OLL2716 or the placebo group (Figure 1). This study was conducted between June 2021 and April 2022 at eight hospitals and clinics (Ageo Central Second Hospital, Saitama, Japan; Kanauchi Medical Clinic, Tokyo, Japan; Kanazawabunko Hospital, Kanagawa, Japan; Fuefuki Central Hospital, Yamanashi, Japan; MY Medical Clinic Shibuya, Tokyo, Japan; Musashisakai Clinic, Tokyo, Japan; Medicaltopia Soka Hospital, Saitama, Japan; and Nihonbashi Sakura Clinic, Tokyo, Japan). The study was approved by the Ethics Committees of Meiji Co., Ltd. (Tokyo, Japan) and Ageo Central Second Hospital (Saitama, Japan). In compliance with the ethical principles of the Declaration of Helsinki and the ethical guidelines for epidemiological studies, the participants were fully informed of the purpose and content of the study, and written informed consent was obtained from all participants before participating in the study. The study protocol was registered on 12 November 2020, using the University Hospital Medical Information Network (UMIN) Clinical Trial Registration System (UMIN000042422). CONSORT 2010 checklist can be found in Appendix A.

### 2.2. Participants

Healthy Japanese adults aged 20–64 years were recruited for this study. Eligibility was determined based on whether participants met the inclusion and exclusion criteria. The inclusion criteria were as follows: (1) healthy Japanese men and women aged 20–64 years at the time of consent; (2) participants who felt “mild” to “slightly severe” postprandial fullness or early satiety on the ‘Individual Gastric Symptom Scores’ at the first screening test and baseline (second screening).

The exclusion criteria were as follows: (1) receiving medical care or treatment for diabetes or gastrointestinal-, eating-, or stress-related psychiatric disorders in the past 6 months; (2) treatment with medication for dyspepsia symptoms in the past 6 months; (3) taking low-dose aspirin or nonsteroidal anti-inflammatory drugs (NSAIDs) in the past 6 months or more; (4) meeting at least one of the first screening or baseline functional dyspepsia Rome IV criteria; (5) serum anti-*H. pylori* antibody titer of ≥10 U/mL at the first *H. pylori* screening test and having received *H. pylori* eradication treatment; (6) severe heartburn or acid reflux at either the first screening or baseline; (7) suspected diabetes mellitus, dyslipidemia, gastrointestinal disease, or severe renal impairment at the first screening or baseline; (8) undergoing any treatment other than those listed previously (excluding transient treatments such as those for the common cold); (9) judged to have a good appetite on the Council on Nutrition Appetite Questionnaire-Japanese (CNAQ-J) at either the first screening or baseline; (10) m-FSSG scores for “postprandial fullness” and “early satiety” were both “0” at baseline; (11) answer was the “best (score 1)” on Q1 of the SF-36v2 at baseline; (12) a total score of 2 on the short-form Nepean Dyspepsia Index (SF-NDI) “irritable, tense, or frustrated because of your stomach problems” at baseline; (13) taking medicines that may affect gastric symptoms for at least 3 days per week for the past one month or more; (14) regular use of foods containing lactic acid bacteria such as yogurt, health foods, or supplements for at least 1 month; (15) taking medicines (such as antibiotics) that affect lactic acid bacteria; (16) smokers; (17) normal alcohol consumption exceeding 40 g per day; (18) food allergies; (19) pregnant, planning or hoping to become pregnant during the study period, or breastfeeding; (20) dental or oral problems that cause bleeding during saliva test; (21) marked lifestyle changes, (22) unable to perform the various testing procedures; (23) would or might travel for more than 1 week during the study period; (24) participated in other clinical trials in the past month or plan to participate in other clinical trials during the study period; and (25) judged by the investigator to be unsuitable for participation. The m-FSSG and SF-36v2 were used only as exclusion criteria during volunteer screening and were not used for the efficacy evaluation. Ultimately, the clinicians made a comprehensive judgment to ensure that no such patients were included in this study.

### 2.3. Study Protocol

The allocation manager randomly assigned participants to two groups, one receiving the test food and the other receiving the control food, using a block randomization method, with age and sex as adjustment factors. The placebo group was asked to ingest 85 g of yogurt per serving, which consisted of raw milk, dairy products, sugar, a sweetener (stevia), and water fermented with *Lactobacillus delbrueckii* subsp. *bulgaricus* and *Streptococcus thermophilus*. The OLL2716 group was asked to ingest yogurt containing *L. paragasseri* OLL2716 (with ≥10^9^ colony-forming units of *L. paragasseri* OLL2716 per serving), added to the same yogurt described for the placebo group. The nutritional values per serving (85 g) of placebo yogurt and yogurt containing *L. paragasseri* OLL2716 were 68 kcal energy, 2.9 g protein, 2.6 g fat, 8.3 g carbohydrate, and 102 mg calcium. After confirming that they were indistinguishable in flavor and appearance, the participants ingested one (85 g) of the assigned test foods per day for 12 weeks. The timing of consumption was not limited. In addition to the person in charge of allocation who assigned the participants to two groups and determined the group of participants, the person who decided the identification number of the test food products and the person who assigned the identification number of the test food products to the group number were separated. As a result, the participants and all study staff were blinded to the test food by using a method in which the three pieces of information necessary for key opening were not collected.

### 2.4. Helicobacter Pylori Screening Test

Serum anti-*H. pylori* antibody titers were measured during the first screening conducted by SRL (Tokyo, Japan), and participants with a titer of 10 U/mL or higher were excluded from the study because they were *H. pylori*-infected.

### 2.5. FD Rome IV Diagnostic Criteria

Participants who answered at least one question regarding postprandial fullness, early satiety, epigastric pain, or epigastric burning at baseline were suspected of having FD using a questionnaire based on the international diagnostic criteria (Rome IV) at first screening and baseline [19]. Two FD subtypes have been defined: (1) postprandial fullness and early satiety, classified as postprandial distress syndrome (PDS); and (2) epigastric pain and burning, classified as epigastric pain syndrome (EPS) [20]. Therefore, participants suspected of having FD symptoms, such as PDS and EPS, based on the Rome IV questionnaire were excluded according to the exclusion criteria, and participants with chronic stomach discomfort were also excluded from this study.

### 2.6. Individual Gastric Symptom Scores (Questionnaire for Gastric Symptoms of the Participants)

A questionnaire on the severity of individual FD and accompanying symptoms was completed during the baseline period and after 6 and 12 weeks of test food intake, as described in a previous study [18]. Participants rated the severity of symptoms (postprandial fullness, early satiety, epigastric bloating, epigastric pain, epigastric burning, heartburn, reflux feeling of gastric acid, nausea, belching, and abdominal bloating) occurring in the prior week on a seven-point Likert scale [18,21,22]: (1) none (absence of symptoms); (2) extremely mild (symptoms could be entirely ignored); (3) mild (symptoms easily tolerated); (4) moderate (symptoms noticed by the patient, but did not affect daily activities); (5) moderate-to-severe (symptoms occasionally limited daily activities); (6) severe (symptoms often limited daily activities); and (7) extremely severe (considerable interference with daily activities, often requiring rest). Furthermore, symptoms including “postprandial fullness” and “early satiety” were categorized as PDS-like symptoms; those including “epigastric pain” and “epigastric burning” as EPS-like symptoms; and symptoms including both PDS- and EPS-like symptoms as FD-like symptoms. This questionnaire has been widely used not only in previous clinical studies on *L. paragasseri* OLL2716 [18] but also as a tool for evaluating upper gastrointestinal symptoms [21,22].

### 2.7. Short-Form Nepean Dyspepsia Index (SF-NDI)

The SF-NDI consists of several items that assess health-related quality of life [23,24]. The questionnaire was administered at the first screening, baseline, and 6 and 12 weeks after consumption of the test foods. Five sub-scales, namely tension (two items, “general emotional well-being” and “irritable, tense, or frustrated”), interference with daily activities [two items, “fun (ability)” and “fun (enjoyment)”], eating/drinking [two items, “eat or drink (ability)” and “eating or drinking (enjoyment)”], knowledge/control [two items, “wondered (always)” and “thought (very serious illness)”], and work/study [two items, “work or study (ability)” and “work or study (enjoyment)”], were derived.

### 2.8. Gastrointestinal Symptom Rating Scale (GSRS)

The questionnaire was administered at baseline and at 6 and 12 weeks after consumption of the test foods. The GSRS is a 15-item, self-administered questionnaire that assesses the severity of a wide range of gastrointestinal symptoms [25,26]; Each of the 15 questions on a Likert scale could be scored on a scale of 1–7, with a total score ranging from 15 to 105 points; however, they were divided by the number of questions to evaluate the score. Five sub-scales, namely reflux syndrome (RS, two items, “heartburn” and “acid regurgitation”), abdominal pain syndrome (AP, three items, “abdominal pain”, “sucking sensations in the epigastrium”, and “nausea and vomiting”), indigestion syndrome (IS, four items, “borborygmus”, “abdominal distension”, “eructation”, and “increased flatus”), constipation syndrome (CS, three items, “decreased passage of stools”, “hard stools”, and “feeling of incomplete evacuation”), and diarrhea syndrome (DS, three items, “increased passage of stools”, “loose stools”, and “urgent need for defecation”), were derived. Furthermore, symptoms including RS, AP, and IS were categorized as upper gastrointestinal (GI)-related, whereas symptoms including CS and DS were categorized as lower GI-related.

### 2.9. Council on Nutrition Appetite Questionnaire-Japanese (CNAQ-J)

The CNAQ-J consists of eight categories: (1) appetite, (2) feeling full, (3) feeling hungry, (4) food tastes, (5) food tastes compared with those when younger, (6) meal frequency per day, (7) feeling sick or nauseated when eating, and (8) usual mood. Each item is answered on a five-point scale ranging from 1 to 5, with the total score ranging from 8 to 40 [27,28]. Additionally, we defined the total score of feeling full and feeling hungry as “pre- and post-meal satisfaction” and the total score of appetite, feeling full, and feeling hungry as “eating satisfaction” for evaluation. The CNAQ-J was administered at the first screening, baseline, and 6 and 12 weeks after consumption of the test foods.

### 2.10. Statistical Analyses

The effect size of *L. paragasseri* OLL2716 in the present study was predicted based on a previous study [18] that examined its effectiveness in improving functional dyspepsia, because no studies have examined the effectiveness of *L. paragasseri* OLL2716 in healthy adults on the improvement of upper gastrointestinal symptoms. In a previous study [18], the disappearance rate of PDS symptoms in individuals with PDS symptoms was 37.5% in the OLL2716 group (*n* = 48) and 17.8% in the placebo group (*n* = 45), with a difference in disappearance rates of 19.7%. In the current study targeting healthy adults with PDS-like symptoms, the difference in disappearance rates between the OLL2716 and placebo groups was 19.7%, similar to that in the previous study [18]. At a two-sided significance level of 5% and power of 80%, the required sample size for each group was calculated to be 78. Considering an approximately 10% dropout rate, the target number of participants in the study was set at 90 for each group.

Intergroup comparisons at baseline were performed using the chi-square test for categorical variables and unpaired Student’s *t*-test for continuous variables. Gastrointestinal symptoms were compared between the two groups using the Wilcoxon signed rank and Fisher’s exact tests. As this study included healthy adults with low-severity upper gastrointestinal symptoms, the improvement rate instead of the disappearance rate was used for the evaluation. To compare the ratio of improvement in gastrointestinal symptoms between the groups, logistic regression analysis with group factors was performed to evaluate the odds ratio, 95% confidence interval, and *p*-value for the placebo group. Data are expressed as means ± standard deviation. All statistical analyses were two-sided, and the significance level was set at 5% with no adjustment for multiple comparisons. Data analysis was conducted using SAS version 9.4 (SAS Institute Inc., Cary, NC, USA) and Bell Curve for Excel 2016 version 3.20 (Social Survey Research Information Co., Ltd., Tokyo, Japan).

## 3. Results

### 3.1. Participant Selection and Baseline Characteristics

A total of 487 Japanese individuals (21–62 years old) who provided written, informed consent were recruited and selected using two screening tests: (1) first screening and (2) second screening (baseline). According to the protocol, after the screening, 174 participants were enrolled in the present study and randomly assigned to the OLL2716 and placebo groups, with 87 participants in each group (Figure 2).

No significant differences were observed in the participants’ background characteristics (Table 1). All 174 participants completed the intake period of 12 weeks and were included in the analysis. No adverse effects or serious adverse events were observed in either group. Additionally, the average intake rate was 99.9% in both the OLL2716 and placebo groups, with the lowest intake rate among the participants being 97.5%. Compliance was confirmed by having the participants record their daily intake in a lifestyle diary.

### 3.2. Primary Endpoint

In the present study, the primary endpoint was the change in gastric symptoms in the participants’ questionnaires, similar to a previous study [18]. Table 2 shows the changes in scores and the number of participants with improved scores. The number of participants with improved scores is further presented as the proportion of the number of participants with improved scores to the number of evaluated participants. The changes in epigastric pain at 6 and 12 weeks were −0.5 ± 1.3 and −0.7 ± 1.2 in the OLL2716 group and −0.1 ± 1.3 and −0.2 ± 1.4 in the placebo group, respectively. The data showed that changes in epigastric pain in the OLL2716 group were significantly improved compared with those in the placebo group at 6 and 12 weeks. The changes in EPS-like symptoms in the OLL2716 group (−0.9 ± 2.1) were significantly improved compared with those in the placebo group (−0.3 ± 2.3) at 6 weeks. Similarly, the number of participants with improved scores for epigastric pain and EPS-like symptoms in the OLL2716 group was 45 and 52, respectively, which was a significant improvement compared with that in the placebo group (30 and 34, respectively) at 6 weeks. No significant differences in other symptoms were noted between the two groups.

### 3.3. Secondary Endpoints

#### 3.3.1. SF-NDI

In the present study, several items concerning the stress caused by gastric symptoms were improved by the intake of the yogurt containing *L. paragasseri* OLL2716 (Table 3). Specifically, the changes in “irritable, tense, or frustrated”, “enjoyment of eating or drinking”, and tension (summary of two items), which are the sub-scales related to mental stress, in the OLL2716 group at 6 weeks were improved compared with those in the placebo group. The number of participants with improved scores for “enjoyment of eating or drinking” in the OLL2716 group was significantly more than that in the placebo group at 6 weeks (Table 3).

#### 3.3.2. GSRS

The changes in the GSRS scores were similar between the two groups (Table 4). The number of participants with improved scores for “lower GI symptoms” and “Over-all” in the OLL2716 group at 12 weeks was significantly more than that in the placebo group. No significant differences in other symptoms were noted between the two groups.

#### 3.3.3. CNAQ-J

Changes in the CNAQ-J scores for feeling hungry at 12 weeks improved with the intake of yogurt containing *L. paragasseri* OLL2716 (Table 5). In addition, changes at 12 weeks in the CNAQ-J scores for “pre- and post-meal satisfaction”, including feeling full and feeling hungry and for “eating satisfaction”, including appetite, feeling full, and feeling hungry, were significantly improved in the OLL2716 group, compared with those in the placebo group. No significant differences in other symptoms were noted between the two groups.

## 4. Discussion

In the present study, we investigated the effects of *L. paragasseri* OLL2716 intake on gastric discomfort and mental stress in healthy Japanese adults with gastric complaints, using four different questionnaires. An evaluation of the questionnaires showed that yogurt containing *L. paragasseri* OLL2716 helped relieve both gastric discomfort and mental stress. At the primary endpoint, epigastric pain and EPS-like symptoms in the OLL2716 group significantly improved compared with those in the placebo group (Table 2). The odds ratios for the improvement in gastric symptoms in the efficacy analysis population were investigated in an additional analysis. The relief odds ratios for the improvement in “epigastric pain” and “EPS-like symptoms” of the OLL2716 and placebo groups after 6 weeks of intake were 2.0 (95% CI, 1.1–3.7, *p* = 0.022) and 2.3 (95% CI, 1.3–4.3, *p* = 0.007), respectively. Similarly, a previous study [15] showed that yogurt containing *L. paragasseri* OLL2716 improved stomach pain caused by an empty stomach. These data strongly suggest that *L. paragasseri* OLL2716 intake reduces the risk of stomach upsets. Interestingly, *L. paragasseri* OLL2716 ameliorated FD in *H. pylori*-uninfected individuals [18]. Therefore, *L. paragasseri* OLL2716 may be a beneficial probiotic strain for the stomach in a wide range of healthy adults and patients. However, because a study on *H. pylori*-uninfected patients with FD showed an improvement in PDS symptoms [18], we hypothesized that PDS-like symptoms would also improve in the present study. The differences in the results could be due to the low baseline values for both PDS- and EPS-like symptoms because this study was conducted on healthy adults. Healthy adults were more responsive than unhealthy adults to changes in EPS-like symptoms because humans are generally more sensitive to pain.

At the secondary endpoint, *L. paragasseri* OLL2716 probably improved the stress associated with gastric disorders, based on the results of the SF-NDI. The SF-NDI is a common questionnaire used to assess the relationship between gastric complaints and the quality of life, including mental stress. In the present study, question items concerning stress, namely “tension” and sub-scales related to mental stress, at 6 weeks, were improved by the intake of the yogurt containing *L. paragasseri* OLL2716. An additional analysis was conducted because the intake of yogurt containing *L. paragasseri* OLL2716 may improve gastrointestinal symptoms and further enhance the quality of life, including mental stress. The relief odds ratios for improvement in “eating or drinking (enjoyment)” and “tension” of the OLL2716 and placebo groups after 6 weeks of intake were 2.0 (95% CI, 1.1–3.6, *p* = 0.031) and 2.0 (95% CI, 1.1–3.7, *p* = 0.022), respectively. The lack of statistical differences at 12 weeks could be due to the beneficial effects of placebo yogurt on the stomach in the long term; however, yogurt containing *L. paragasseri* OLL2716 was considered to be more effective earlier. To the best of our knowledge, these are the first subjective results indicating a relationship between the stomach and stress in *L. paragasseri* OLL2716-ingesting participants. Thus, *L. paragasseri* OLL2716 may ameliorate both gastric discomfort and mental stress in healthy adults with gastric complaints.

The GSRS is a popular questionnaire that is used to assess gastrointestinal symptoms. Whereas the change in GSRS scores was similar between the groups, the number of “Over-all” improved participants (total assessment) in the OLL2716 group at 12 weeks was improved compared with the placebo group. Compared with the placebo group, the odds ratios for improvement in “lower GI symptom” and “Over-all” in the OLL2716 group after 12 weeks of intake were 1.9 (95% CI, 1.0–3.5, *p* = 0.035) and 2.0 (95% CI, 1.1–3.8, *p* = 0.029), respectively. The results of the primary endpoint and GSRS evaluations suggest that *L. paragasseri* OLL2716 has a positive effect on the entire digestive tract. Otomi et al. reported that *L. paragasseri* OLL2716 improved GSRS scores [15] and suggested that *L. paragasseri* OLL2716 helped relieve upper gastrointestinal symptoms. However, the current study suggests that *L. paragasseri* OLL2716 alleviates upper and lower gastrointestinal symptoms. Further research is required to clarify whether *L. paragasseri* OLL2716 alleviates the lower gastrointestinal symptoms.

The CNAQ-J is commonly used to assess appetite. The changes at 12 weeks in the CNAQ-J score of feeling hungry, “pre- and post-meal satisfaction” and “eating satisfaction”, were improved by the intake of the yogurt containing *L. paragasseri* OLL2716. Gastrointestinal signaling, such as neural pathways and released peptide hormones, has been reported to influence the feelings of satiety and hunger [29]. Additionally, acute mental stress suppresses appetite [30,31]. Therefore, in this study, it was considered possible that improvements in the gastrointestinal symptoms and stress led to the improvement in appetite.

The efficacy of *L. paragasseri* OLL2716 was not limited to gastrointestinal symptoms such as stomach pain and EPS-like symptoms but also affected stress and appetite. These improvements were considered to be due to the consumption of yogurt containing *L. paragasseri* OLL2716, which alleviated gastrointestinal symptoms such as stomach pain and subsequently led to reduced stress and improved appetite.

All previous clinical studies on *L. paragasseri* OLL2716 [15,18,32,33,34] were conducted with live bacteria in yogurt, leading us to consider that *L. paragasseri* OLL2716 is most effective when included in yogurt in the live bacterial state. Therefore, we chose yogurt as the formulation and set yogurt as the placebo. However, the control food was yogurt containing *Lactobacillus delbrueckii* subsp. *bulgaricus* and *Streptococcus thermophilus*, known as starter lactic acid bacteria, which are necessary for preparing yogurt. Therefore, considering that yogurt itself has a beneficial effect on the digestive tract, these effects may lead to an improvement in gastric symptoms. Additionally, yogurt is primarily composed of milk ingredients and contains α-lactalbumin, one of the whey proteins, and casein, both of which have been reported to have analgesic effects [35,36]. Given that the placebo yogurt contained these proteins in addition to the two types of lactic acid bacteria, it is possible that the placebo yogurt also demonstrated a certain degree of improvement. Nonetheless, it is considered to be highly significant that the continued intake of *L. paragasseri* OLL2716 resulted in even greater improvement. 

We identified three limitations of this study: an unclear mechanism of action, the selection of participants primarily based on subjective measures, and the observation period. One limitation of this study is that although we found that the regular intake of yogurt containing *L. paragasseri* OLL2716 improved both gastric discomfort and mental stress in healthy adults with gastric complaints, the mechanism of action remains unclear. One of the presumed mechanisms for the beneficial effects of *L. paragasseri* OLL2716 is thought to be associated with the regulation of autonomic function because previous studies [3,15] have reported the improvement of stress markers, such as immunoglobulin A and amylase concentrations in saliva, also known as sympathetic markers. Moreover, in a previous study on *L. paragasseri* OLL2716, the main effects were observed in PDS symptoms [18]; however, in the present study, improvements were also observed in EPS-like symptoms such as stomach pain. Stomach pain is associated with peptic ulcers caused by stomach acid or *H. pylori* infection as well as gastritis caused due to medications such as NSAIDs, infections, alcohol, and gastroesophageal reflux disease [37,38,39,40]. The present study targeted healthy adults, excluding those with *H. pylori* infections, medication use, or excessive alcohol consumption; therefore, these factors were not thought to be the cause of stomach pain. Imbalances in autonomic nervous system function have also been reported as the causes of stomach pain [41]. Given that previous studies [3,15] have reported improvements in stress and sympathetic markers, the effects of *L. paragasseri* OLL2716 on the autonomic nervous system may also contribute to the improvement of stomach pain. However, as evaluations to support this finding were not conducted in the present study, further evidence of EPS-like symptoms is required. 

Second, the selection of participants primarily based on subjective measures may have resulted in an inability to completely exclude participants with conditions such as gastroparesis, which could have prevented us from clearly demonstrating the differences between the OLL2716 and placebo groups. In this study, it was not feasible to conduct gastric emptying tests for all the participants because of scheduling constraints. However, in future studies, conducting such tests on selected participants may help clarify participant characteristics. 

Regarding the third limitation, an increasing number of participants showed improvement after 6 and 12 weeks. Therefore, extending the intervention period might have led to more pronounced improvement effects. Future research should consider the duration of intake based on the evaluation criteria.

As suggested by this study, continuous intake of *L. paragasseri* OLL2716 may help improve gastric symptoms and alleviate stress, which in turn suggests that functional foods containing *L. paragasseri* OLL2716 could enhance the quality of life in healthy adults. Although the continuous intake of *L. paragasseri* OLL2716 has been reported to improve symptoms in patients with FD [18] and in those infected with *H. pylori* [32], its impact on the quality of life in these patients has not been investigated. Therefore, it is important to explore whether continuous intake of *L. paragasseri* OLL2716 could also contribute to improving the quality of life associated with symptoms in these patients, making this a topic for further research.

## 5. Conclusions

In conclusion, our findings suggest that regular intake of *L. paragasseri* OLL2716 may improve both gastric discomfort and mental stress in healthy adults with gastric complaints such as postprandial fullness or early satiety. Although one possible mechanism is the regulation of autonomic function, further research is required to clarify the mechanism of action of *L. paragasseri* OLL2716.

## 6. Patents

N.Y., K.K., A.N., and H.K. are the inventors of pending patents (Japanese Patent Application No. 2023–015847).

## Figures and Tables

**Figure 1 nutrients-16-03188-f001:**
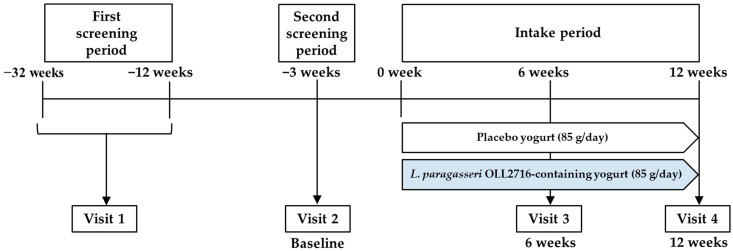
Study design and schedule to investigate the effects of *L. paragasseri* OLL2716 on gastric discomfort and mental stress in healthy adults with gastric complaints.

**Figure 2 nutrients-16-03188-f002:**
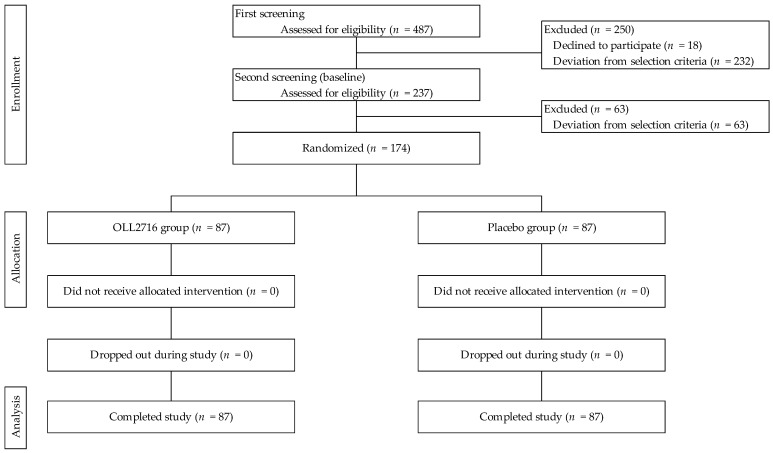
Flow chart of participants in this study.

**Table 1 nutrients-16-03188-t001:** Baseline characteristics of the participants.

Characteristics	OLL2716 (*n* = 87)	Placebo (*n* = 87)	*p*-Value
Female	55	56	0.875 ^1^
Male	32	31	
Age (year)	37.4 ± 11.9	37.0 ± 11.1	0.792 ^2^
Postprandial fullness score	2.5 ± 0.7	2.5 ± 0.8	0.839 ^2^
Early satiety score	2.1 ± 0.9	2.1 ± 1.0	0.814 ^2^
BMI [kg(m^2^)^−1^]	22.2 ± 3.7	22.4 ± 3.5	0.777 ^2^

Data are shown as mean ± standard deviation. Intergroup comparison: ^1^ Chi-square test, ^2^
*t*-test; BMI, body mass index.

**Table 2 nutrients-16-03188-t002:** Score change before ingestion and the number of participants with improved scores in the Individual Gastric Symptom Scores.

Items					Score Change before Ingestion (⊿)				Number of Participants with Improved Scores (%)					
	Group	0 Week	6 Weeks	12 Weeks	6 Weeks–0 Week	*p* Value ^1^	12 Weeks–0 Week	*p* Value ^1^	6 Weeks		*p* Value ^2^	12 Weeks		*p* Value ^2^
1. Postprandial fullness	OLL2716	2.5 ± 0.7	2.0 ± 0.8	1.7 ± 0.9	−0.5 ± 0.8	0.564	−0.8 ± 1.0	0.745	37	(42.5%)	0.879	50	(57.5%)	1.000
	Placebo	2.5 ± 0.8	1.9 ± 0.9	1.6 ± 1.0	−0.6 ± 1.1		−0.9 ± 1.2		39	(44.8%)		49	(56.3%)	
2. Early satiety	OLL2716	2.1 ± 0.9	1.9 ± 1.0	1.6 ± 1.1	−0.2 ± 1.0	0.483	−0.5 ± 1.0	0.841	29	(33.3%)	0.528	37	(42.5%)	0.760
	Placebo	2.1 ± 1.0	1.8 ± 1.0	1.6 ± 1.1	−0.3 ± 1.0		−0.5 ± 1.1		34	(39.1%)		40	(46.0%)	
3. Epigastric bloating	OLL2716	2.3 ± 1.1	1.9 ± 1.0	1.6 ± 1.1	−0.5 ± 1.1	0.362	−0.7 ± 1.1	0.861	46	(52.9%)	0.544	48	(55.2%)	1.000
	Placebo	2.2 ± 1.2	1.9 ± 1.3	1.5 ± 1.1	−0.4 ± 1.2		−0.7 ± 1.4		41	(47.1%)		48	(55.2%)	
4. Epigastric pain	OLL2716	1.6 ± 1.1	1.1 ± 1.1	1.0 ± 1.0	−0.5 ± 1.3	0.020	−0.7 ± 1.2	0.028	45	(51.7%)	0.032	47	(54.0%)	0.095
	Placebo	1.3 ± 1.2	1.2 ± 1.1	1.1 ± 1.0	−0.1 ± 1.3		−0.2 ± 1.4		30	(34.5%)		35	(40.2%)	
5. Epigastric burning	OLL2716	1.3 ± 1.2	0.9 ± 1.0	1.0 ± 1.0	−0.4 ± 1.0	0.256	−0.4 ± 1.0	0.289	33	(37.9%)	0.753	34	(39.1%)	0.754
	Placebo	1.1 ± 1.2	0.9 ± 0.9	0.9 ± 1.0	−0.3 ± 1.3		−0.2 ± 1.4		30	(34.5%)		31	(35.6%)	
6. Heartburn	OLL2716	1.5 ± 1.2	1.1 ± 1.0	1.0 ± 1.0	−0.5 ± 1.1	0.550	−0.6 ± 1.1	0.322	37	(42.5%)	1.000	50	(57.5%)	0.225
	Placebo	1.4 ± 1.2	1.1 ± 1.1	1.0 ± 0.9	−0.3 ± 1.2		−0.4 ± 1.3		36	(41.4%)		41	(47.1%)	
7. Reflex feeling of gastric acid	OLL2716	1.5 ± 1.1	1.0 ± 1.0	1.0 ± 1.0	−0.5 ± 1.0	0.258	−0.5 ± 1.1	0.094	41	(47.1%)	0.166	39	(44.8%)	0.442
	Placebo	1.2 ± 1.2	0.9 ± 1.0	1.0 ± 1.0	−0.3 ± 1.2		−0.2 ± 1.3		31	(35.6%)		33	(37.9%)	
8. Nausea	OLL2716	1.0 ± 1.1	0.7 ± 1.0	0.7 ± 1.0	−0.3 ± 1.0	0.583	−0.3 ± 0.9	0.413	29	(33.3%)	0.873	30	(34.5%)	1.000
	Placebo	1.0 ± 1.0	0.8 ± 1.0	0.8 ± 0.9	−0.2 ± 1.0		−0.2 ± 1.2		31	(35.6%)		30	(34.5%)	
9. Belching	OLL2716	1.6 ± 1.2	1.4 ± 1.1	1.2 ± 1.1	−0.3 ± 1.1	0.653	−0.4 ± 1.0	0.818	33	(37.9%)	0.877	40	(46.0%)	1.000
	Placebo	1.7 ± 1.3	1.3 ± 1.1	1.3 ± 1.0	−0.3 ± 1.1		−0.4 ± 1.2		35	(40.2%)		40	(46.0%)	
10. Abdominal bloating	OLL2716	2.1 ± 1.2	1.7 ± 1.0	1.6 ± 1.2	−0.4 ± 1.2	0.108	−0.5 ± 1.3	0.638	43	(49.4%)	0.091	41	(47.1%)	0.649
	Placebo	2.0 ± 1.4	1.7 ± 1.3	1.4 ± 1.1	−0.2 ± 1.3		−0.6 ± 1.4		31	(35.6%)		45	(51.7%)	
PDS-like (1 and 2)	OLL2716	4.5 ± 1.3	3.8 ± 1.6	3.3 ± 1.8	−0.7 ± 1.5	0.455	−1.2 ± 1.7	0.689	43	(49.4%)	0.880	51	(58.6%)	0.757
	Placebo	4.6 ± 1.4	3.7 ± 1.8	3.2 ± 1.9	−0.9 ± 1.7		−1.4 ± 2.0		45	(51.7%)		54	(62.1%)	
EPS-like (4 and 5)	OLL2716	3.0 ± 2.1	2.0 ± 1.9	1.9 ± 1.9	−0.9 ± 2.1	0.028	−1.0 ± 1.9	0.073	52	(59.8%)	0.010	49	(56.3%)	0.068
	Placebo	2.4 ± 2.0	2.1 ± 1.8	1.9 ± 1.8	−0.3 ± 2.3		−0.5 ± 2.5		34	(39.1%)		36	(41.4%)	
FD-like (PDS-like and EPS-like)	OLL2716	7.5 ± 2.9	5.9 ± 3.0	5.2 ± 3.3	−1.6 ± 3.0	0.207	−2.3 ± 3.0	0.377	50	(57.5%)	0.129	52	(59.8%)	0.540
	Placebo	7.0 ± 2.8	5.7 ± 3.1	5.1 ± 3.4	−1.3 ± 3.3		−1.9 ± 4.0		39	(44.8%)		47	(54.0%)	

Changes in scores and number of participants with improved scores from baseline. Data are shown as mean ± standard deviation (*n* = 87 in the OLL2716 group and *n* = 87 in the placebo group). Intergroup comparison: ^1^ Wilcoxon rank sum test and ^2^ Fisher’s exact test. Number of improved participants (%): (number of participants who improved/total number of participants evaluated) × 100. PDS, postprandial distress syndrome; EPS, epigastric pain syndrome; FD, functional dyspepsia. Sub-scales: PDS-like, “postprandial fullness” and “early satiety”; EPS-like, “epigastric pain” and “epigastric burning”; and FD-like, PDS- and EPS-like.

**Table 3 nutrients-16-03188-t003:** Score change before ingestion and the number of participants with improved scores in the Short-form Nepean Dyspepsia Index (SF-NDI).

Items					Score Change before Ingestion (⊿)				Number of Participants with Improved Scores (%)			
	Group	0 Week	6 Weeks	12 Weeks	6 Weeks–0 Week	*p* Value ^1^	12 Weeks–0 Week	*p* Value^1^	6 Weeks	*p* Value ^2^	12 Weeks	*p* Value ^2^
1. General emotional well-being	OLL2716	2.0 ± 0.6	1.7 ± 0.6	1.6 ± 0.6	−0.3 ± 0.7	0.084	−0.4 ± 0.7	0.822	28 (32.2%)	0.121	39 (44.8%)	0.760
	Placebo	2.0 ± 0.6	1.9 ± 0.6	1.6 ± 0.5	−0.1 ± 0.7		−0.4 ± 0.7		18 (20.7%)		36 (41.4%)	
2. Irritable, tense, or frustrated	OLL2716	2.2 ± 0.5	1.7 ± 0.6	1.6 ± 0.6	−0.5 ± 0.6	0.042	−0.6 ± 0.7	0.666	40 (46.0%)	0.061	47 (54.0%)	0.544
	Placebo	2.2 ± 0.5	1.9 ± 0.6	1.6 ± 0.6	−0.3 ± 0.7		−0.6 ± 0.7		27 (31.0%)		42 (48.3%)	
3. Fun (ability)	OLL2716	1.7 ± 0.7	1.3 ± 0.5	1.3 ± 0.5	−0.3 ± 0.8	0.353	−0.4 ± 0.8	0.457	34 (39.1%)	0.429	39 (44.8%)	0.279
	Placebo	1.6 ± 0.8	1.4 ± 0.6	1.3 ± 0.5	−0.2 ± 0.7		−0.3 ± 0.7		28 (32.2%)		31 (35.6%)	
4. Fun (enjoyment)	OLL2716	1.7 ± 0.7	1.3 ± 0.5	1.3 ± 0.5	−0.4 ± 0.8	0.284	−0.4 ± 0.8	0.909	38 (44.2%)	0.279	39 (45.3%)	0.760
	Placebo	1.7 ± 0.8	1.4 ± 0.7	1.3 ± 0.5	−0.3 ± 0.8		−0.4 ± 0.7		31 (35.6%)		37 (42.5%)	
5. Eat or drink (ability)	OLL2716	1.9 ± 0.7	1.6 ± 0.6	1.4 ± 0.6	−0.2 ± 0.7	0.215	−0.5 ± 0.7	0.226	29 (33.3%)	0.177	44 (50.6%)	0.287
	Placebo	1.8 ± 0.7	1.6 ± 0.6	1.4 ± 0.6	−0.1 ± 0.6		−0.4 ± 0.7		20 (23.0%)		36 (41.4%)	
6. Eating or drinking (enjoyment)	OLL2716	2.0 ± 0.7	1.5 ± 0.6	1.4 ± 0.6	−0.5 ± 0.9	0.031	−0.6 ± 0.8	0.150	42 (48.3%)	0.044	52 (59.8%)	0.171
	Placebo	1.9 ± 0.8	1.6 ± 0.6	1.4 ± 0.6	−0.2 ± 0.7		−0.5 ± 0.8		28 (32.2%)		42 (48.3%)	
7. Wondered (always)	OLL2716	1.7 ± 0.7	1.3 ± 0.5	1.2 ± 0.5	−0.4 ± 0.7	0.421	−0.4 ± 0.8	0.454	31 (35.6%)	0.63	38 (43.7%)	0.537
	Placebo	1.6 ± 0.8	1.4 ± 0.6	1.2 ± 0.5	−0.3 ± 0.8		−0.4 ± 0.9		27 (31.0%)		33 (37.9%)	
8. Thought (very serious illness)	OLL2716	1.3 ± 0.5	1.1 ± 0.4	1.1 ± 0.4	−0.1 ± 0.5	0.182	−0.2 ± 0.5	0.097	16 (18.4%)	0.403	19 (21.8%)	0.328
	Placebo	1.2 ± 0.4	1.2 ± 0.4	1.1 ± 0.3	0.0 ± 0.5		−0.1 ± 0.5		11 (12.6%)		13 (14.9%)	
9. Work or study (ability)	OLL2716	1.6 ± 0.6	1.5 ± 0.6	1.3 ± 0.5	−0.1 ± 0.7	0.620	−0.3 ± 0.7	0.677	24 (27.6%)	1.000	30 (34.5%)	1.000
	Placebo	1.6 ± 0.6	1.4 ± 0.7	1.3 ± 0.5	−0.1 ± 0.7		−0.3 ± 0.6		24 (27.6%)		29 (33.3%)	
10. Work or study (enjoyment)	OLL2716	1.6 ± 0.7	1.4 ± 0.6	1.3 ± 0.6	−0.2 ± 0.8	0.681	−0.3 ± 0.8	0.890	30 (34.5%)	0.747	31 (35.6%)	0.749
	Placebo	1.6 ± 0.7	1.4 ± 0.6	1.3 ± 0.5	−0.2 ± 0.8		−0.3 ± 0.7		27 (31.0%)		28 (32.2%)	
Tension (1 and 2)	OLL2716	4.1 ± 0.9	3.4 ± 1.1	3.1 ± 1.2	−0.7 ± 1.1	0.026	−1.0 ± 1.2	0.675	45 (51.7%)	0.032	54 (62.1%)	0.357
	Placebo	4.2 ± 1.0	3.8 ± 1.1	3.2 ± 1.1	−0.4 ± 1.3		−1.0 ± 1.2		30 (34.5%)		47 (54.0%)	
Interference with daily activities (3 and 4)	OLL2716	3.4 ± 1.2	2.7 ± 1.0	2.6 ± 1.0	−0.7 ± 1.4	0.262	−0.8 ± 1.4	0.575	42 (48.8%)	0.361	46 (53.5%)	0.288
	Placebo	3.3 ± 1.5	2.8 ± 1.3	2.5 ± 1.0	−0.5 ± 1.4		−0.8 ± 1.3		36 (41.4%)		39 (44.8%)	
Eating/drinking (5 and 6)	OLL2716	3.9 ± 1.3	3.2 ± 1.0	2.8 ± 1.1	−0.7 ± 1.4	0.057	−1.1 ± 1.5	0.095	45 (51.7%)	0.224	56 (64.4%)	0.217
	Placebo	3.6 ± 1.3	3.2 ± 1.1	2.8 ± 1.1	−0.4 ± 1.1		−0.9 ± 1.3		36 (41.4%)		47 (54.0%)	
Knowledge/control (7 and 8)	OLL2716	3.0 ± 1.0	2.5 ± 0.8	2.4 ± 0.8	−0.5 ± 1.0	0.240	−0.6 ± 1.1	0.223	37 (42.5%)	0.210	41 (47.1%)	0.358
	Placebo	2.9 ± 1.2	2.5 ± 0.9	2.4 ± 0.8	−0.3 ± 1.1		−0.5 ± 1.3		28 (32.2%)		34 (39.1%)	
Work/study (9 and 10)	OLL2716	3.2 ± 1.3	2.8 ± 1.1	2.6 ± 1.0	−0.3 ± 1.5	0.977	−0.6 ± 1.4	0.942	33 (37.9%)	0.875	36 (41.4%)	1.000
	Placebo	3.2 ± 1.3	2.9 ± 1.3	2.6 ± 0.9	−0.3 ± 1.4		−0.7 ± 1.2		31 (35.6%)		35 (40.2%)	

Changes in scores and number of participants with improved scores from baseline. Data are shown as mean ± standard deviation (*n* = 87 in the OLL2716 group and *n* = 87 in the placebo group). Intergroup comparison: ^1^ Wilcoxon rank sum test and ^2^ Fisher’s exact test. Number of improved participants (%): (number of participants who improved/total number of participants evaluated) × 100. Sub-scales: tension, “general emotional well-being” and “irritable, tense, or frustrated”; interference with daily activities, “fun (ability)” and “fun (enjoyment)”; eating/drinking, “eat or drink (ability)” and “eating or drinking (enjoyment)”; knowledge/control, “wondered (always)” and “thought (very serious illness)”; and work/study, “work or study (ability)” and “work or study (enjoyment)”.

**Table 4 nutrients-16-03188-t004:** Score change before ingestion and the number of participants with improved scores in the Gastrointestinal Symptom Rating Scale (GSRS).

Items					Score Change before Ingestion (⊿)				Number of Participants with Improved Scores (%)			
	Group	0 Week	6 Weeks	12 Weeks	6 Weeks–0 Week	*p* Value ^1^	12 Weeks–0 Week	*p* Value ^1^	6 Weeks	*p* Value ^2^	12 Weeks	*p* Value ^2^
1. Abdominal pain	OLL2716	2.5 ± 1.1	2.0 ± 0.8	1.9 ± 0.8	−0.4 ± 1.2	0.369	−0.6 ± 1.1	0.143	37 (42.5%)	0.643	48 (55.2%)	0.288
	Placebo	2.2 ± 1.0	2.0 ± 1.0	1.8 ± 1.0	−0.2 ± 1.1		−0.4 ± 1.1		33 (37.9%)		40 (46.0%)	
2. Heartburn	OLL2716	2.1 ± 0.8	1.9 ± 0.8	1.7 ± 0.8	−0.2 ± 1.0	0.439	−0.4 ± 0.8	0.641	27 (31.0%)	0.522	38 (43.7%)	0.879
	Placebo	2.1 ± 1.0	1.8 ± 0.9	1.6 ± 0.8	−0.3 ± 1.0		−0.5 ± 0.9		32 (36.8%)		40 (46.0%)	
3. Acid regurgitation	OLL2716	2 ± 0.9	1.8 ± 0.8	1.6 ± 0.7	−0.2 ± 0.9	0.843	−0.4 ± 0.8	0.220	26 (29.9%)	0.736	42 (48.3%)	0.221
	Placebo	1.9 ± 0.9	1.7 ± 0.8	1.6 ± 0.8	−0.2 ± 0.9		−0.3 ± 1.0		23 (26.4%)		33 (37.9%)	
4. Sucking sensations in the epigastrium	OLL2716	2.1 ± 0.9	1.9 ± 0.9	1.7 ± 0.8	−0.2 ± 1.0	0.534	−0.4 ± 1.0	0.228	34 (39.1%)	0.340	38 (43.7%)	0.440
	Placebo	2.0 ± 0.8	1.8 ± 0.9	1.7 ± 0.9	−0.2 ± 0.9		−0.2 ± 0.9		27 (31.0%)		32 (36.8%)	
5. Nausea and vomiting	OLL2716	1.9 ± 1.0	1.6 ± 0.9	1.6 ± 0.7	−0.3 ± 1.2	0.498	−0.3 ± 0.8	0.701	33 (37.9%)	0.144	33 (37.9%)	0.635
	Placebo	1.9 ± 0.9	1.8 ± 0.8	1.7 ± 0.9	−0.1 ± 0.9		−0.3 ± 0.9		23 (26.4%)		29 (33.3%)	
6. Borborygmus	OLL2716	2.3 ± 1.1	2.4 ± 1.0	2.1 ± 1.0	0.0 ± 1.0	0.089	−0.3 ± 1.0	0.931	24 (27.6%)	0.147	34 (39.1%)	0.754
	Placebo	2.6 ± 1.2	2.3 ± 1.1	2.2 ± 1.1	−0.3 ± 1.1		−0.3 ± 1.2		34 (39.1%)		31 (35.6%)	
7. Abdominal distension	OLL2716	2.4 ± 1.0	2.1 ± 0.8	2.0 ± 0.8	−0.3 ± 1.1	0.507	−0.4 ± 1.0	0.654	35 (40.2%)	1.000	38 (43.7%)	0.761
	Placebo	2.4 ± 1.0	2.1 ± 0.9	1.9 ± 0.9	−0.4 ± 1.0		−0.5 ± 1.1		34 (39.1%)		41 (47.1%)	
8. Eructation	OLL2716	2.0 ± 0.9	1.8 ± 0.8	1.8 ± 0.9	−0.2 ± 0.9	0.671	−0.2 ± 0.9	0.830	27 (31.0%)	0.522	29 (33.3%)	0.528
	Placebo	2.1 ± 1.1	1.9 ± 0.8	1.8 ± 0.8	−0.3 ± 1.1		−0.3 ± 1.0		32 (36.8%)		34 (39.1%)	
9. Increased flatus	OLL2716	2.4 ± 1.0	2.2 ± 0.9	2.0 ± 0.8	−0.2 ± 1.0	0.821	−0.4 ± 1.0	0.605	32 (36.8%)	0.751	39 (44.8%)	0.646
	Placebo	2.5 ± 1.2	2.3 ± 1.1	2.1 ± 0.9	−0.2 ± 1.1		−0.4 ± 1.0		29 (33.3%)		35 (40.2%)	
10. Decreased passage of stools	OLL2716	2.4 ± 1.1	2.1 ± 1.0	1.8 ± 0.9	−0.3 ± 1.0	0.479	−0.6 ± 1.1	0.526	33 (37.9%)	0.877	42 (48.3%)	0.543
	Placebo	2.4 ± 1.4	2.0 ± 1.2	1.9 ± 1.0	−0.4 ± 1.1		−0.5 ± 1.2		35 (40.2%)		37 (42.5%)	
11. Increased passage of stools	OLL2716	2.1 ± 1.1	1.9 ± 1.0	1.6 ± 0.7	−0.1 ± 1.0	0.532	−0.4 ± 0.9	0.147	27 (31.0%)	0.747	34 (39.1%)	0.264
	Placebo	1.9 ± 1.1	1.7 ± 1.1	1.7 ± 1.0	−0.2 ± 1.0		−0.2 ± 1.1		30 (34.5%)		26 (29.9%)	
12. Loose stools	OLL2716	2.0 ± 1.0	1.9 ± 0.9	1.7 ± 0.7	−0.1 ± 0.9	0.813	−0.3 ± 0.8	0.617	26 (29.9%)	1.000	29 (33.3%)	1.000
	Placebo	1.9 ± 1.1	1.7 ± 0.8	1.6 ± 0.8	−0.2 ± 1.1		−0.3 ± 1.1		25 (28.7%)		28 (32.2%)	
13. Hard stools	OLL2716	2.2 ± 1.1	2.1 ± 1.0	1.8 ± 0.8	−0.1 ± 0.9	0.273	−0.4 ± 1.0	0.827	29 (33.3%)	0.347	33 (37.9%)	0.757
	Placebo	2.3 ± 1.2	1.9 ± 1.0	1.9 ± 1.0	−0.3 ± 1.1		−0.4 ± 1.2		36 (41.4%)		36 (41.4%)	
14. Urgent need for defecation	OLL2716	2.2 ± 1.0	2.2 ± 1.0	1.8 ± 0.8	0.0 ± 1.1	0.598	−0.4 ± 1.0	0.151	25 (28.7%)	1.000	34 (39.1%)	0.637
	Placebo	2.2 ± 1.2	2.2 ± 1.1	2.1 ± 1.1	0.0 ± 1.1		−0.1 ± 1.2		26 (29.9%)		30 (34.5%)	
15. Feeling of incomplete evacuation	OLL2716	2.4 ± 1.1	2.2 ± 0.9	1.9 ± 0.9	−0.2 ± 1.1	0.510	−0.5 ± 1.0	0.225	31 (35.6%)	0.630	41 (47.1%)	0.358
	Placebo	2.3 ± 1.0	2.2 ± 1.0	2.0 ± 0.9	−0.1 ± 0.9		−0.3 ± 1.0		27 (31.0%)		34 (39.1%)	
RS (2 and 3)	OLL2716	2.1 ± 0.8	1.9 ± 0.7	1.7 ± 0.7	−0.2 ± 0.8	0.780	−0.4 ± 0.7	0.762	31 (35.6%)	0.639	50 (57.5%)	0.648
	Placebo	2.0 ± 0.8	1.8 ± 0.7	1.6 ± 0.7	−0.3 ± 0.9		−0.4 ± 0.9		35 (40.2%)		46 (52.9%)	
AP (1, 4, and 5)	OLL2716	2.1 ± 0.8	1.8 ± 0.8	1.7 ± 0.7	−0.3 ± 0.9	0.251	−0.5 ± 0.8	0.126	44 (50.6%)	0.362	54 (62.1%)	0.169
	Placebo	2.0 ± 0.8	1.9 ± 0.8	1.7 ± 0.8	−0.2 ± 0.7		−0.3 ± 0.8		37 (42.5%)		44 (50.6%)	
IS (6, 7, 8, and 9)	OLL2716	2.3 ± 0.8	2.1 ± 0.7	2.0 ± 0.7	−0.2 ± 0.8	0.370	−0.3 ± 0.8	0.859	32 (36.8%)	0.219	48 (55.2%)	0.363
	Placebo	2.4 ± 0.9	2.1 ± 0.8	2.0 ± 0.8	−0.3 ± 0.8		−0.4 ± 0.9		41 (47.1%)		41 (47.1%)	
CS (10, 13, and 15)	OLL2716	2.4 ± 0.9	2.1 ± 0.8	1.8 ± 0.8	−0.2 ± 0.8	0.775	−0.5 ± 0.8	0.227	41 (47.1%)	0.879	54 (62.1%)	0.221
	Placebo	2.3 ± 1.0	2.1 ± 1.0	1.9 ± 0.9	−0.3 ± 0.8		−0.4 ± 0.9		43 (49.4%)		45 (51.7%)	
DS (11, 12, and 14)	OLL2716	2.1 ± 0.9	2.0 ± 0.8	1.7 ± 0.7	−0.1 ± 0.8	0.638	−0.4 ± 0.7	0.127	32 (36.8%)	1.000	45 (51.7%)	0.288
	Placebo	2.0 ± 1.0	1.9 ± 0.8	1.8 ± 0.8	−0.1 ± 0.7		−0.2 ± 0.9		31 (35.6%)		37 (42.5%)	
Upper GI (RS, AP, and IS)	OLL2716	2.2 ± 0.7	2.0 ± 0.6	1.8 ± 0.6	−0.2 ± 0.7	0.970	−0.4 ± 0.7	0.424	28 (32.2%)	0.271	47 (54.0%)	0.225
	Placebo	2.2 ± 0.8	2.0 ± 0.7	1.8 ± 0.7	−0.2 ± 0.7		−0.4 ± 0.7		36 (41.4%)		38 (43.7%)	
Lower GI (CS and DS)	OLL2716	2.2 ± 0.8	2.1 ± 0.7	1.8 ± 0.6	−0.1 ± 0.7	0.730	−0.4 ± 0.7	0.078	36 (41.4%)	0.533	52 (59.8%)	0.048
	Placebo	2.2 ± 0.9	2.0 ± 0.7	1.9 ± 0.7	−0.2 ± 0.6		−0.3 ± 0.8		31 (35.6%)		38 (43.7%)	
Over-all	OLL2716	2.2 ± 0.6	2.0 ± 0.6	1.8 ± 0.6	−0.2 ± 0.7	0.855	−0.4 ± 0.6	0.160	19 (21.8%)	0.229	39 (44.8%)	0.041
	Placebo	2.2 ± 0.7	2.0 ± 0.6	1.8 ± 0.7	−0.2 ± 0.6		−0.3 ± 0.7		27 (31.0%)		25 (28.7%)	

Changes in scores and number of participants with improved scores from baseline. Data are shown as mean ± standard deviation (*n* = 87 in the OLL2716 group and *n* = 87 in the placebo group). Intergroup comparison: ^1^ Wilcoxon rank sum test and ^2^ Fisher’s exact test. Number of improved participants (%): (number of participants who improved/total number of participants evaluated) × 100. RS, reflux syndrome; AP, abdominal pain syndrome; IS, indigestion syndrome; CS, constipation syndrome; DS, diarrhea syndrome; GI, gastrointestinal. Sub-scales: RS, “heartburn” and “acid regurgitation”; AP, “abdominal pain”, “sucking sensations in the epigastrium”, and “nausea and vomiting”; IS, “borborygmus”, “abdominal distension”, “eructation”, and “increased flatus”; CS, “decreased passage of stools”, “hard stools”, and “feeling of incomplete evacuation”; DS, “increased passage of stools”, “loose stools”, and “urgent need for defecation”; Upper GI, RS, AP, and IS; Lower GI, CS and DS.

**Table 5 nutrients-16-03188-t005:** Score change before ingestion and the number of participants with improved scores in the Council on Nutrition Appetite Questionnaire-Japanese (CNAQ-J).

Items					Score Change before Ingestion (⊿)				Number of Participants with Improved Scores (%)					
	Group	0 Week	6 Weeks	12 Weeks	6 Weeks–0 Week	*p* Value ^1^	12 Weeks–0 Week	*p* Value ^1^	6 Weeks		*p* Value ^2^	12 Weeks		*p* Value ^2^
1. Appetite	OLL2716	3.3 ± 0.5	3.4 ± 0.5	3.4 ± 0.6	0.1 ± 0.5	0.393	0.1 ± 0.6	0.207	6	(6.9%)	0.307	7	(8.0%)	0.331
	Placebo	3.3 ± 0.6	3.4 ± 0.6	3.4 ± 0.6	0.0 ± 0.6		0.1 ± 0.6		11	(12.6%)		12	(13.8%)	
2. Feeling full	OLL2716	3.7 ± 0.5	3.7 ± 0.4	3.8 ± 0.4	0.1 ± 0.4	0.502	0.1 ± 0.5	0.781	5	(5.7%)	0.188	4	(4.6%)	0.535
	Placebo	3.7 ± 0.5	3.7 ± 0.4	3.8 ± 0.4	0.0 ± 0.5		0.1 ± 0.5		11	(12.6%)		7	(8.0%)	
3. Feeling hungry	OLL2716	3.0 ± 0.7	3.0 ± 0.8	3.1 ± 0.8	0.1 ± 0.7	0.570	0.1 ± 0.8	0.020	13	(14.9%)	0.685	13	(14.9%)	0.063
	Placebo	3.1 ± 0.8	3.2 ± 0.7	3.0 ± 0.7	0.0 ± 0.7		–0.1 ± 0.8		16	(18.4%)		24	(27.6%)	
4. Food tastes	OLL2716	3.6 ± 0.6	3.7 ± 0.6	3.7 ± 0.7	0.1 ± 0.6	0.271	0.1 ± 0.6	0.330	6	(6.9%)	0.432	9	(10.3%)	0.194
	Placebo	3.7 ± 0.6	3.7 ± 0.6	3.8 ± 0.7	0.1 ± 0.6		0.1 ± 0.7		10	(11.5%)		16	(18.4%)	
5. Food tastes compared to when younger	OLL2716	3.2 ± 0.6	3.3 ± 0.6	3.2 ± 0.6	0.0 ± 0.6	0.538	0.0 ± 0.7	0.873	9	(10.3%)	1.000	13	(14.9%)	1.000
	Placebo	3.3 ± 0.6	3.4 ± 0.7	3.3 ± 0.6	0.1 ± 0.6		0.0 ± 0.6		9	(10.3%)		12	(13.8%)	
6. Meal frequency a day	OLL2716	3.8 ± 0.4	3.8 ± 0.4	3.9 ± 0.4	0.0 ± 0.3	0.314	0.0 ± 0.4	0.162	3	(3.4%)	0.720	2	(2.3%)	1.000
	Placebo	3.9 ± 0.3	3.9 ± 0.4	3.9 ± 0.3	0.0 ± 0.4		0.0 ± 0.2		5	(5.7%)		2	(2.3%)	
7. Feeling sick or nauseated when eating	OLL2716	4.1 ± 0.7	4.3 ± 0.7	4.4 ± 0.7	0.2 ± 0.7	0.367	0.3 ± 0.7	0.210	11	(12.6%)	0.524	9	(10.3%)	0.194
	Placebo	4.2 ± 0.7	4.3 ± 0.7	4.3 ± 0.7	0.1 ± 0.9		0.1 ± 0.9		15	(17.2%)		16	(18,4%)	
8. Usual mood	OLL2716	3.4 ± 0.6	3.4 ± 0.6	3.4 ± 0.6	0.0 ± 0.5	0.087	0.0 ± 0.5	0.070	9	(10.3%)	0.794	7	(10.3%)	0.794
	Placebo	3.3 ± 0.5	3.4 ± 0.6	3.4 ± 0.5	0.1 ± 0.5		0.1 ± 0.5		7	(8.0%)		7	(8.0%)	
2 and 3	OLL2716	6.7 ± 0.9	6.8 ± 0.9	6.9 ± 1.0	0.1 ± 0.8	0.603	0.3 ± 1.0	0.045		–	–		–	–
	Placebo	6.9 ± 0.9	6.9 ± 0.8	6.9 ± 0.8	0.0 ± 0.9		0.0 ± 1.0			–	–		–	–
1, 2, and 3	OLL2716	9.9 ± 1.2	10.1 ± 1.2	10.3 ± 1.3	0.2 ± 1.0	0.531	0.4 ± 1.3	0.022		–	–		–	–
	Placebo	10.2 ± 1.3	10.3 ± 1.1	10.3 ± 1.1	0.1 ± 1.2		0.0 ± 1.3			–	–		–	–
Over-all	OLL2716	28.1 ± 2.5	28.7 ± 2.3	29.0 ± 2.5	0.6 ± 2.0	0.711	0.9 ± 2.4	0.144		–	–		–	–
	Placebo	28.6 ± 2.4	29.0 ± 2.5	29.0 ± 2.3	0.4 ± 2.2		0.4 ± 2.3			–	–		–	–

Changes in scores and number of participants with improved scores from baseline. Data are shown as mean ± standard deviation (*n* = 87 in the OLL2716 group and *n* = 87 in the placebo group). Intergroup comparison: ^1^ Wilcoxon rank sum test and ^2^ Fisher’s exact test. Number of improved participants (%): (number of participants who improved/total number of participants evaluated) × 100. Sub-scales: pre- and post-meal satisfaction, “feeling full” and “feeling hungry”; eating satisfaction, “appetite”, “feeling full”, and “feeling hungry”.

## Data Availability

Data supporting the results of this study are available from the corresponding author upon reasonable request. However, data for the test food analysis result are not publicly available due to include confidential product specifications.

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
