# Peer review of "The Beneficial Effects of Regular Intake of Lactobacillus paragasseri OLL2716 on Gastric Discomfort in Healthy Adults: A Randomized, Double-Blind, Placebo-Controlled Study"

_nutrients, 2024, doi:10.3390/nu16183188_

Round 1

Reviewer 1 Report

Comments and Suggestions for Authors

1) The authors should discuss what these observations mean to a general audience, and how these results should be interpreted and how they may affect future medical practice.

2) A limitation that should be discuss is that the study is based only on self-reporting questionnaires with no clinicians involved to validate the symptoms on otherwise healthy subjects.

3) Provide manufacturing date and lot numbers for both probiotic and placebo. Give certificate of analysis for both probiotic and placebo as supplementary files. Give material safety data sheet for both probiotic and placebo as supplementary files. These all files are necessary to confirm and validate quality of the said formula used in the investigation.

4) Did the researchers have the subjects return empty test articles?  How was compliance assessed/confirmed?

5) The large number of individuals excluded from participation based on the criteria provided suggests that these observations may not be relatable to an average individual.

Comments on the Quality of English Language

Only minor edits necessary, for example, page 4 line 149-151 states "as described in a previous study" without a citation.

Author Response

We thank the reviewer for careful reading our manuscript and for giving useful comments. Our responses to the reviewers' reports are as follows. The added or revised sections in the manuscript are indicated in bold and underlined. In addition to the revisions made in response to the comments from reviewers, there are also some changes that do not alter the content. In the revised version, the sections that have been modified from the previous manuscript are highlighted.

Comments 1: The authors should discuss what these observations mean to a general audience, and how these results should be interpreted and how they may affect future medical practice.

Response 1: We wish to thank the reviewer for this suggestion. We have added the following to the discussion (Page 15, lines 434-442 in the revised version): “As suggested by this study, continuous intake of L. paragasseri OLL2716 may help improve gastric symptoms and alleviate stress, which in turn suggests that functional foods containing L. paragasseri OLL2716 could enhance the quality of life in healthy adults. Although the continuous intake of L. paragasseri OLL2716 has been reported to improve symptoms in patients with FD [18] and in those infected with H. pylori [32], its impact on the quality of life in these patients has not been investigated. Therefore, it is important to explore whether continuous intake of L. paragasseri OLL2716 could also contribute to improving the quality of life associated with symptoms in these patients, making this a topic for further research.

Comments 2: A limitation that should be discuss is that the study is based only on self-reporting questionnaires with no clinicians involved to validate the symptoms on otherwise healthy subjects. The authors should discuss what these observations mean to a general audience, and how these results should be interpreted and how they may affect future medical practice.

Response 2: We wish to thank the reviewer for this comment. We objectively assessed the presence of H. pylori infection and excluded participants who were infected. Additionally, we used the Rome IV questionnaire to exclude subjects suspected of having FD. Ultimately, the clinicians made a comprehensive judgment to ensure that no patients were included in the study. We have added to the ‘participants’ section that the clinicians, based on the exclusion criteria, made a comprehensive judgment to ensure that no patients were included. In our future research, we would consider not only subjective evaluations but also objective evaluations. We have added the following to the Materials and Methods (Page 3, lines 114-115 in the revised version): “Ultimately, the clinicians made a comprehensive judgment to ensure that no such pa-tients were included in this study.”.

In our future research, we will consider not only subjective evaluations but also objective evaluations.

Comments 3: Provide manufacturing date and lot numbers for both probiotic and placebo. Give certificate of analysis for both probiotic and placebo as supplementary files. Give material safety data sheet for both probiotic and placebo as supplementary files. These all files are necessary to confirm and validate quality of the said formula used in the investigation.

Response 3: We wish to thank the reviewer for this comment. Since the test food was yogurt and a chilled product, we sent the test food to participants once a week during the intake period. We are attaching files ‘test food analysis result management sheet’ and ‘test food analysis result management sheet’ here, but since they contain confidential information such as product planning values, we would prefer not to include them as supplementary files for the paper if possible. Could this be arranged?

Comments 4: Did the researchers have the subjects return empty test articles?  How was compliance assessed/confirmed?

Response 4: We wish to thank the reviewer for this comment. Since the test food was yogurt and a chilled product, we sent the test food to participants once a week during the intake period. Due to operational reasons, we did not collect empty containers. Compliance was confirmed by having participants record their daily intake in a lifestyle diary. The intake rate was 99.9% for both the OLL2716 group and the control group, with the lowest intake rate among participants being 97.5%.

We have added the following to the Results (Page 6, lines 247-249 in the revised version): “… group. Additionally, the average intake rate was 99.9% in both the OLL2716 and placebo groups, with the lowest intake rate among the participants being 97.5%. Compliance was confirmed by having the participants record their daily intake in a lifestyle diary.”.

Comments 5: The large number of individuals excluded from participation based on the criteria provided suggests that these observations may not be relatable to an average individual.

Response 5: We wish to thank the reviewer for this comment. In this study, we established inclusion and exclusion criteria to target healthy individuals with gastric discomfort. As you pointed out, there were many participants who did not fully meet the inclusion or exclusion criteria. Therefore, although the study initially focused on healthy adults, we have revised clarifications in the abstract and conclusions to specify that the study targets healthy individuals with gastric discomfort, even though they are not classified as patients.

We have added the following to the Abstract (Page1, lines 28-30 in the revised version): “The study findings suggest that regular intake of L. paragasseri OLL2716 may improve both gastric discomfort and mental stress in healthy adults with gastric complaints such as postprandial fullness or early satiety.” and the Conclusions (Page 15, lines 444-446 in the revised version): “In conclusion, our findings suggest that regular intake of L. paragasseri OLL2716 may improve both gastric discomfort and mental stress in healthy adults with gastric complaints such as postprandial fullness or early satiety.”.

Comments on the Quality of English Language :

Only minor edits necessary, for example, page 4 line 149-151 states "as described in a previous study" without a citation.

Response : We wish to thank the reviewer for this comment. We have added citation as follows (Page 4, lines 154-156 in the revised version): “A questionnaire on the severity of individual FD and accompanying symptoms was completed during the baseline period and after 6 and 12 weeks of test food intake, as described in a previous study [18].”.

Additionally, I corrected instances where citations from previous studies were missing.

Reviewer 2 Report

Comments and Suggestions for Authors

Dear authors 

I read with very great interest your RCT on Lactobacillus paragasseri in patients with gastrointestinal symptoms and although very interesting there are some concerns I'd like to raise

The major issue relies in the inclusion criteria of the study which in my opinion introduce a relevant selection bias in the population, affecting the overall results: you included healthy patients (which are not indeed healthy since even not fulfilling the Rome IV criteria for PDS or EPS they indeed had to show typical FD symptom namely post-prandial fullness or early satiety) with gastric fullness or early satiety which consist of 2 of the main symptoms of FD and even better of gastroparesis. The absence of a gastric emptying test to exclude GP patients among the two arms of this RCT could have hindered considerably the results especially in the placebo group. 

Furthermore the design of the study in itself deserves some considerations: the follow-up of the study is somehow short only 12 weeks (probably for ethical issues?). Probably extending the follow-up timing could ameliorate the outcomes.

Moreover, very few metrics (for clinical scores as epigastric pain or EPS-like bloating or SF-NDI, as for GSRS), showed quite weak significances with raw data not so distant from the PBO group (did not get what statistical adjustment and compensation tests were used to calculate those). The GSRS scale and SF-NDI which are by far the most used in clinical practice among the one listed did not show almost any significant changes. There are clinical scores with higher responsiveness and more adequate recall period to exclude recall biases (24 hours) like the LPDS, FDSD, mSDA and other validated useful questionnaires used in multiplem studies like the SF36, PAGI-QoL, PAGI-SYM and even the GCSI (Smeets FGM et al. Neurogastroenterol Motil. 2018  doi: 10.1111/nmo.13327.). In addition the NDI reliability and validity have not been assessed for symptoms. This is a point which could be discussed. It's fairly hard to draw conclusion on the effectiveness of lactobacilli from these results. Furthermore stating a clear success or improvement definition it would be interesting.

Additionally also the "potential" role of the placebo yogurt formulation on the placebo arms outcomes should be discussed and could be of interest.

Side considerations: concerning sample size calculation based on the power of 80% to detect a 19.7% a difference disappearing rate of PDS symptoms based on one only previous study published seems weak. Could you be more precise and prediction for between-groups clinical scores, absolute changes for example? In outcomes assessment could you be more precise in stating what tests were used to calculate between-arms differences and adjusted for whta factors (covariance analysis) to strenghten your considerations? 

Thank you for the opportunity to analyze this work   

Comments on the Quality of English Language

Minor editing

Author Response

We thank the reviewer for careful reading our manuscript and for giving useful comments. Our responses to the reviewers' reports are as follows. The added or revised sections in the manuscript are indicated in bold and underlined. In addition to the revisions made in response to the comments from reviewers, there are also some changes that do not alter the content. In the revised version, the sections that have been modified from the previous manuscript are highlighted.

Comments 1: The major issue relies in the inclusion criteria of the study which in my opinion introduce a relevant selection bias in the population, affecting the overall results: you included healthy patients (which are not indeed healthy since even not fulfilling the Rome IV criteria for PDS or EPS they indeed had to show typical FD symptom namely post-prandial fullness or early satiety) with gastric fullness or early satiety which consist of 2 of the main symptoms of FD and even better of gastroparesis. The absence of a gastric emptying test to exclude GP patients among the two arms of this RCT could have hindered considerably the results especially in the placebo group.

Furthermore, the design of the study in itself deserves some considerations: the follow-up of the study is somehow short only 12 weeks (probably for ethical issues?). Probably extending the follow-up timing could ameliorate the outcomes.

Response 1: We wish to thank the reviewer for this comment. In the exclusion criteria, we used the Rome IV questionnaire to exclude participants suspected of having FD. However, the final decision was made by the clinicians, who ensured that no patients were included in the study.

Additionally, no participants required medical visits for symptoms of postprandial fullness or early satiety during the intervention period, we believe that this study targeted participants with gastric discomfort within the range of normal health. As you pointed out, we did not conduct a gastric emptying test, which could have excluded gastroparesis and potentially led to a more appropriate selection of participants. Moreover, since our study observed symptom improvement after 6 and 12 weeks of intake, extending the observation period might have resulted in further improvement. The subjective evaluations during participant selection and these other aspects are considered limitations of this study, and we have added them to the limitations section of the discussion. We have added the following to the Discussion (Page 14, lines 401-402 in the revised version): “ We identified three limitations of this study: an unclear mechanism of action, selection of participants primarily based on subjective measures, and the observation period.”, (Page 14-15, lines 423-433 in the revised version): “Second, the selection of participants primarily based on subjective measures may have resulted in an inability to completely exclude participants with conditions such as gastroparesis, which could have prevented us from clearly demonstrating the differences between the OLL2716 and placebo groups. In this study, it was not feasible to conduct gastric emptying tests for all the participants because of scheduling constraints. However, in future studies, conducting such tests on selected participants may help clarify participant characteristics.

Regarding the third limitation, an increasing number of participants showed improvement after 6 and 12 weeks. Therefore, extending the intervention period might have led to more pronounced improvement effects. Future research should consider the duration of intake based on the evaluation criteria.”.

Comments 2: Moreover, very few metrics (for clinical scores as epigastric pain or EPS-like bloating or SF-NDI, as for GSRS), showed quite weak significances with raw data not so distant from the PBO group (did not get what statistical adjustment and compensation tests were used to calculate those). The GSRS scale and SF-NDI which are by far the most used in clinical practice among the one listed did not show almost any significant changes. There are clinical scores with higher responsiveness and more adequate recall period to exclude recall biases (24 hours) like the LPDS, FDSD, mSDA and other validated useful questionnaires used in multiplem studies like the SF36, PAGI-QoL, PAGI-SYM and even the GCSI (Smeets FGM et al. Neurogastroenterol Motil. 2018 doi: 10.1111/nmo.13327.). In addition the NDI reliability and validity have not been assessed for symptoms. This is a point which could be discussed. It's fairly hard to draw conclusion on the effectiveness of lactobacilli from these results. Furthermore stating a clear success or improvement definition it would be interesting.

Response 2: We wish to thank the reviewer for this comment. In the statistical analysis for this study, gastrointestinal symptoms were compared between the two groups using the Wilcoxon signed-rank and Fisher’s exact tests. Therefore, we did not use any statistical analysis methods that required adjustment. For the evaluation methods in this study, we adopted ‘Individual Gastric Symptom Scores’, which has been used in previous trials evaluating gastric symptoms involving L. paragasseri OLL2716, as well as GSRS and SF-NDI, which are authorized methods. Additionally, since we believe that the effects of continuous intake of food may manifest over a longer period, we considered it preferable to evaluate over a slightly longer duration of around one week rather than a short 24-hour period. Furthermore, to assess the impact on stress related to gastric symptoms, we decided to include SF-NDI.

Thank you for providing information on many validated questionnaires. We will consider them in our future research.

Comments 3: Additionally also the "potential" role of the placebo yogurt formulation on the placebo arms outcomes should be discussed and could be of interest.

Response 3: We wish to thank the reviewer for this comment. We chose yogurt as the formulation because we considered that L. paragasseri OLL2716 would be most effective when included in yogurt in a live bacterial state, and we set the placebo as yogurt as well. However, the placebo was also yogurt and contained two types of lactic acid bacteria, known as starters, necessary for preparing yogurt. Therefore, considering that yogurt itself has a beneficial effect on the digestive tract, it is expected that these effects may lead to an improvement in gastric symptoms. Additionally, yogurt is primarily composed of milk ingredients, and contains α-lactalbumin, one of the whey proteins, and casein, both of which have been reported to have analgesic effects. Given that the placebo yogurt contains these proteins in addition to the two types of lactic acid bacteria, it is possible that the placebo also demonstrated a certain degree of improvement. The test food was the placebo yogurt with L. paragasseri OLL2716 added. We have added a discussion regarding the potential improvement effects of the placebo yogurt. (Page 14, lines 387-400 in the revised version): “All previous clinical studies on L. paragasseri OLL2716 [15,18,32-34] were con-ducted with live bacteria in yogurt, leading us to consider that L. paragasseri OLL2716 is most effective when included in yogurt in the live bacterial state. Therefore, we chose yogurt as the formulation and set yogurt as the placebo. However, the control food was yogurt containing Lactobacillus delbrueckii subsp. bulgaricus and Streptococcus thermophilus, known as starter lactic acid bacteria, which are necessary for preparing yogurt. Therefore, considering that yogurt itself has a beneficial effect on the digestive tract, these effects may lead to an improvement in gastric symptoms. Additionally, yogurt is primarily composed of milk ingredients, and contains α-lactalbumin, one of the whey proteins, and casein, both of which have been reported to have analgesic effects [35,36]. Given that the placebo yogurt contained these proteins in addition to the two types of lactic acid bacteria, it is possible that the placebo yogurt also demonstrated a certain degree of improvement. Nonetheless, it is considered to be highly significant that continued intake of L. paragasseri OLL2716 resulted in even greater improvement.”.

Comments 4: Side considerations: concerning sample size calculation based on the power of 80% to detect a 19.7% a difference disappearing rate of PDS symptoms based on one only previous study published seems weak. Could you be more precise and prediction for between-groups clinical scores, absolute changes for example? In outcomes assessment could you be more precise in stating what tests were used to calculate between-arms differences and adjusted for whta factors (covariance analysis) to strenghten your considerations? Thank you for the opportunity to analyze this work.

Response 4: We wish to thank the reviewer for this comment. The only clinical trial available for reference in determining the sample size was the one we referred to for this study, making it difficult to make further predictions. Additionally, in the statistical analysis for this study, gastrointestinal symptoms were compared between the two groups using the Wilcoxon signed-rank and Fisher’s exact tests. Therefore, we did not use any statistical analysis methods that required adjustment.

Reviewer 3 Report

Comments and Suggestions for Authors

The manuscript describes beneficial effect of regular intake of Lactobacillus paragasseri OLL2716 on gastric discomfort in healthy adults: A randomized, double-blind, placebo-controlled study. The topic is relevant to the aim and scope of the Nutrients. The manuscript is well written and easy to follow. However, this manuscript falls short of acceptance criteria due to the comments below:

Since the ranges of all data have overlapped between OLL2716 and placebo tremendously, the findings of this study little support that L. paragasseri OLL2716 improves both gastric discomfort and mental stress in healthy adults with gastric complaints.

Author Response

We thank the reviewer for careful reading our manuscript and for giving useful comments. Our responses to the reviewers' reports are as follows. The added or revised sections in the manuscript are indicated in bold and underlined. In addition to the revisions made in response to the comments from reviewers, there are also some changes that do not alter the content. In the revised version, the sections that have been modified from the previous manuscript are highlighted.

Comments 1: Since the ranges of all data have overlapped between OLL2716 and placebo tremendously, the findings of this study little support that L. paragasseri OLL2716 improves both gastric discomfort and mental stress in healthy adults with gastric complaints.

Response 1: We wish to thank the reviewer for this comment. We chose yogurt as the formulation because we believed that L. paragasseri OLL2716 would be most effective when included in yogurt in a live bacterial state, and we set the placebo as yogurt as well. However, the placebo was also yogurt and contained two types of lactic acid bacteria, known as starters, necessary for preparing yogurt. Therefore, considering that yogurt itself has a beneficial effect on the digestive tract, it is expected that these effects may lead to an improvement in gastric symptoms. Additionally, yogurt is primarily composed of milk ingredients, and contains α-lactalbumin, one of the whey proteins, and casein, both of which have been reported to have analgesic effects. Given that the placebo yogurt contains these proteins in addition to the two types of lactic acid bacteria, it is possible that the placebo group also demonstrated a certain degree of improvement. The test food was the placebo yogurt with L. paragasseri OLL2716 added. We have added a discussion regarding the potential improvement effects of the placebo yogurt. Therefore, since yogurt itself may have had some improvement effects, the differences between the two groups may have been limited to only a few items. Given that the differences between the two groups were found in only a few items, we have revised the conclusion as follows (Page 15, lines 444-446 in the revised version): “In conclusion, our findings suggest that regular intake of L. paragasseri OLL2716 may improve both gastric discomfort and mental stress in healthy adults with gastric complaints such as postprandial fullness or early satiety.”.

Round 2

Reviewer 3 Report

Comments and Suggestions for Authors

The issue is addressed.